# Cracking Resistance of Steam-Cured Precast Concrete Using High Alite Cement with Modified Fly Ash

**Aghiad Alhafez [1,\*], Shingo Miyazawa [2,\*], Nobukazu Nito [3], Ryuichiroh Kuga [4] and Etsuo Sakai [5]**

1. Graduate School of Engineering, Ashikaga University, Ashikaga 326-8558, Japan
2. Department of Civil Engineering, Ashikaga University, Ashikaga 326-8558, Japan
3. Technical Department, DC Co., Ltd., Kawasaki 210-0854, Japan; nito_nobukazu@dccorp.jp
4. Central Research Laboratory, Taiheiyo Cement Corporation, Sakura 285-8655, Japan; ryuichiroh_kuga@taiheiyo-cement.co.jp
5. Tokyo Institute of Technology (Professor Emeritus), Tokyo 152-8552, Japan; etsusaka.honjo@gmail.com
* Correspondence: aghiad979@hotmail.com (A.A.); miyazawa.shingo@g.ashikaga.ac.jp (S.M.)

**Abstract:** Cement with fly ash has rarely been used in Japan, mainly because its strength development is slower than ordinary Portland cement. In this research, the effect of the new type of fly ash (which was modified by the electrostatic belt separation method) with high alite ($C_3S$) cement on cracking resistance of precast concrete prepared by steam curing was studied. The mechanical and shrinkage properties of the proposed fly ash concrete were compared with those of concrete made using OPC cement without fly ash. In order to study the cracking tendency of precast concrete with the proposed cement with fly ash, thermal stress analysis was conducted, taking into consideration the experimental data of concrete properties with the different concrete mix proportions. A standard precast concrete box culvert model was used in this 3D FEM analysis, and the distribution of temperature and relative humidity in the cross-section and induced restraint stress during and after steam curing were discussed. Steam-cured concrete with fly ash and high alite cement developed higher compressive strength on the first day of age than concrete with OPC. The proposed fly ash concrete developed high cracking resistance in the early days. On the other hand, the results showed that the drying shrinkage at later ages was the main cause of cracking.

**Keywords:** fly ash; precast concrete; compressive strength; steam curing; high alite cement; 3D FEM analysis; drying shrinkage; cracking resistance

## 1. Introduction

Fly ash is a residual material of energy production that is driven out of the boiler with the flue gases and has numerous advantages for use in concrete [1]. Due to rising landfill costs and increased interest in sustainable development, recycling fly ash has become a severe problem in recent years—it is one of the primary sources of carbon dioxide emissions and air pollution on Earth. However, it is nevertheless seen as a green resource, and it is mainly regarded as eco-friendly when utilized in buildings because it is a recycled material. Considering that power plants will burn coal and produce fly ash regardless, in that case, it will be very beneficial to use it to save money and energy in the construction sector.

The use of FA as an admixture at concrete-mixing plants is still rare in Japan. This is mainly because the strength development of FA cement concrete is slower than that of OPC concrete. Another reason is that the unburned carbon content in fly ash affects the properties of concrete and makes quality control very difficult. To solve this problem, a new modified fly ash with low unburned carbon has been developed using an electrostatic belt separation method, which is a special technique using the differences in conductivity between the particles of the minerals to separate them from each other [2,3].

Precast concrete is widely used globally for many kinds of construction. Additionally, in Japan, in order to cope with the declining birthrate and aging population, the use of

precast concrete products is increasing year by year because it is effective in labor-saving construction.

One of the most important points to focus on in any industry is increasing productivity, and in the precast concrete industry, we need to use the molds as many times as possible in the shortest time. This means demolding in the shortest possible time; therefore, a suitable compressive strength must be reached to demold safely. The most common way to improve the compressive strength of precast concrete at an early age is by using the steam-curing method [4,5], which influences the physical properties of the concrete. However, using fly ash will slow down the compressive strength development of the concrete, as mentioned before. To solve this problem, it is recommended to use high alite ($C_3S$) cement with modified fly ash to accelerate the strength development of the concrete [6]. In general, the main chemical components of OPC are alite ($C_3S$) and belite ($C_2S$). Alite's chemical reaction is much faster than belite's; as a result, it is responsible for the early strength of concrete [7]. Supported by the results of a previous study [8], the authors reported that using high alite cement with fly ash will increase the compressive strength at early ages in cast-in-place concrete structures.

Controlling concrete cracking, which can appear in precast concrete products at various ages due to the material's quasi-brittle nature and limited ability to deform under tensile stresses [9–11], is one of the critical problems. Concrete shrinkage is frequently blamed for cracks that may appear sooner or later, and the size of those cracks has an adverse impact on the durability of the concrete. It has been acknowledged that the correct material selection, mix proportions, curing conditions, and construction techniques can control concrete cracking brought on by both temperature change and shrinkage [12].

In this research, the effect of the new fly ash, modified by the electrostatic belt separation method with high alite ($C_3S$) cement, on precast concrete prepared by steam curing was studied. The mechanical and shrinkage properties of the proposed fly ash concrete were compared with those of concrete made using ordinary Portland cement without fly ash. In order to study the cracking tendency of precast concrete with the proposed cement with fly ash, thermal stress analysis was conducted, taking into consideration the experimental data of concrete properties with the different concrete mix proportions. A standard precast concrete box culvert model was used in this 3D FEM analysis, and temperature distribution in the cross-section and induced restraint stress during and after steam curing were discussed.

## 2. Materials and Mix Proportions

Table 1 shows the physical and chemical properties of the binder materials, high alite cement (A), and ordinary Portland cement (OPC (N)), which were used for the experiments in this study.

**Table 1.** Physical and chemical properties of cement.

| Name | Density (g/cm³) | Blaine Fineness (cm²/g) | f.CaO (%) | Mineral Composition (%) | | | |
|---|---|---|---|---|---|---|---|
| | | | | $C_3S$ | $C_2S$ | $C_3A$ | $C_4AF$ |
| A | 3.11 | 5380 | 2.1 | 69.3 | 2.9 | 9.4 | 7.7 |
| N | 3.16 | 3170 | 0.2 | 61.6 | 15.1 | 8.2 | 9.1 |

(A) high alite cement and (N) ordinary Portland cement (OPC).

Ordinary Portland cement is commercially produced and conforms to the Japanese standard JIS R 5210: 2009—standard specification for Portland cement [13].

The Blaine fineness is higher and the amount of free calcium oxide (f.CaO) and alite ($C_3S$) in the high alite cement (A) are larger than those in conventional ordinary Portland cement (N); however, the production process in the cement factories for both is almost the same regarding the raw materials, kiln temperature, and power consumption for raw materials' treatment [8]. For that reason, it can be said, from the environmental viewpoint,

that the emission of $CO_2$ by using fly ash with high alite cement is less than when producing OPC.

Table 2 shows the physical and chemical properties of the fly ash used for experiments in this study. FA-1 is the raw material for FA-2. FA-2 is modified fly ash from which unburnt carbon is removed from FA-1 by the electrostatic belt separation method. FA in Table 2, which is also modified fly ash prepared by the same method, was used for the experiments indicated in the following sections. As shown in the table, the ignition loss and methylene blue adsorption of FA-2 and FA are much lower than that of FA-1, suggesting that unburnt carbon is effectively removed. It can also be said from Table 2 that the strength activity index of FA-2 is higher than that of FA-1 and that the reactivity of fly ash is improved by the treatment using the electrostatic belt separation method. The samples of fly ash are by-products from a coal-fired power plant and can be categorized to Class F fly ash specified by ASTM C618-05 [14]. The properties of these samples conform to the requirements of fly ash Class II specified by JIS A 6201 [6], except that the ignition loss of FA-1 is higher than the upper limit of 5.0%. The Blaine fineness values of the samples, which are controlled by using an air separator, are equivalent to those of commonly used fly ash for concrete in Japan.

**Table 2.** Physical and chemical properties of fly ash.

| Name | Density (g/cm$^3$) | Blaine Fineness (cm$^2$/g) | SiO$_2$ (%) | Ig. Loss (%) | Flow Percent of Control (%) | Methylene Blue Adsorption (mg/g) | Strength Activity Index (%) | | |
|------|--------|--------|--------|--------|--------|--------|--------|--------|--------|
| | | | | | | | 7 Days | 28 Days | 91 Days |
| FA-1 | 2.24 | 3900 | 59.7 | 5.2 | 101 | 1.96 | 74 | 82 | 92 |
| FA-2 | 2.25 | 4030 | 63.9 | 0.8 | 106 | 0.53 | 78 | 85 | 97 |
| FA | 2.19 | 3490 | 65.3 | 0.9 | 106 | 0.33 | 74 | 84 | 99 |

Table 3 shows the four mix proportions of concrete used in the experiments. The fly ash referred to as "FA" in Table 2 was used for these concrete mixtures, but FA-1 and FA-2 were not used for the concrete experiments. An air-entraining agent (AE) and superplasticizer (SP) were used to introduce entrained air and improve the workability. The target slump and air content for the experiments were $12 \pm 2.5$ cm and $4.5 \pm 1.0\%$, respectively. The replacement ratio of the fly ash in the cement was taken to be 18%.

**Table 3.** Mix proportions of concrete.

| Proportion | W/B (%) | s/a (%) | Amounts of Contents (kg/m$^3$) | | | | | | Chemical Admixture (B X%) | | |
|------|------|------|------|------|------|------|------|------|------|------|------|
| | | | Water | N | A | FA | S | G | SP | AE303 | AE785 |
| N 45% | 45 | 45 | 160 | 356 | ... | ... | 795 | 993 | 0.8 | 0.001 | ... |
| A + FA 45% | | 45 | 160 | ... | 292 | 64 | 783 | 977 | 0.75 | ... | 0.03 |
| N 33% | 33 | 43 | 160 | 485 | ... | ... | 715 | 967 | 0.88 | 0.0015 | ... |
| A + FA 33% | | 43 | 160 | ... | 398 | 87 | 699 | 946 | 0.8 | ... | 0.035 |

(W) Water, (B) binder, (S) sand, (G) coarse aggregates, and (a) volume of aggregate.

The coarse aggregate used in this study was crushed sandstone obtained from Kuzu region in Japan, while river sand obtained from Kinugawa River in Japan was used as the fine aggregate. Table 4 shows the physical properties of the aggregates that were used in this study.

**Table 4.** Physical properties of the aggregates.

| Aggregates | Provenance | Maximum Size (mm) | Density (g/cm$^3$) | Absorption (%) | Fineness Modulus |
|---|---|---|---|---|---|
| Coarse aggregates (a) | Sandstone from Kuzu region | 20 | 2.62 | 0.76 | . . . |
| Fine aggregates (sand) | River sand from Kinugawa River | . . . | 2.60 | 2.11 | 2.75 |

## 3. Experimental Procedures

### 3.1. Compressive Strength and Modulus of Elasticity

For the compressive strength tests, cylindrical concrete specimens of 100 mm in diameter and 200 mm in height were cast according to JIS A1108: 2006—method of test for compressive strength of concrete [15], and ASTM C39/C39M-18: 2018—standard test method for compressive strength of cylindrical concrete specimens [16].

After casting, the specimens for steam curing were stored in a temperature-controlled room at 20 °C and 80% R.H. After 2 h of casting time, the specimens for 1, 14, and 91 days of the compressive strength test were subjected to steam curing. The temperature gradually increased during 2 h and 15 min, up to 65 °C as a stable maximum temperature for 3 h, and the target temperature used in the experiments was similar to steam curing in general precast concrete factories. The temperature gradually reduced during 10 h to stable at 20 °C for 6 h and 45 min, the total time for the steam-curing procedure was 24 h. Figure 1 shows the general temperature regime for a typical steam-curing profile.

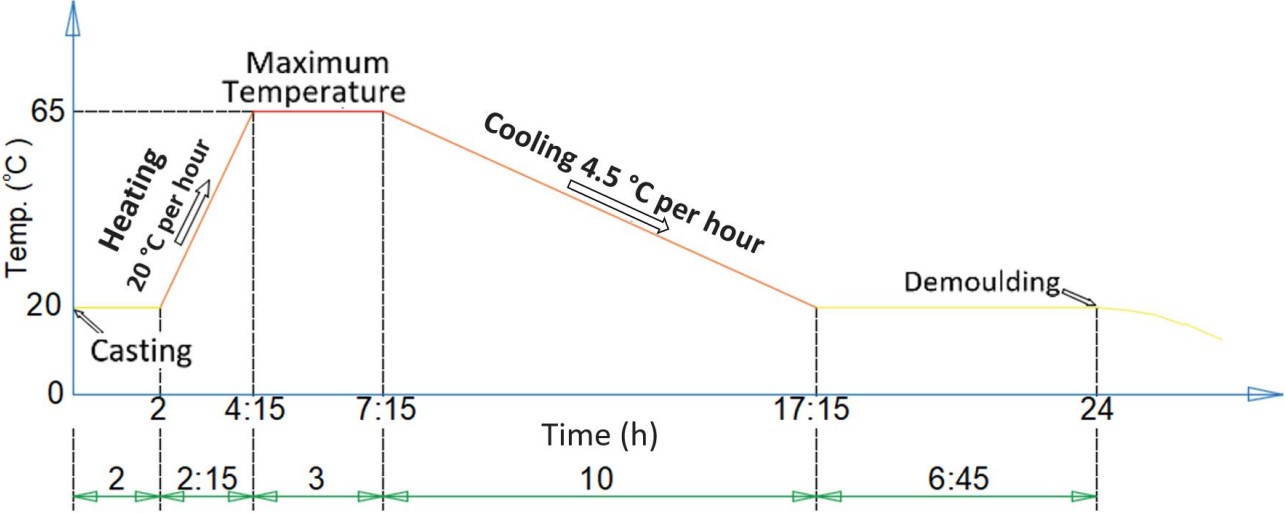

**Figure 1.** Steam-curing temperature history.

After steam curing, the specimens were demolded after 24 h and the compressive strength was measured. Then, the steam-cured specimens were subjected to drying conditions in a temperature-controlled room at 20 °C and 60% R.H. to observe the compressive strength at 14 and 91 days.

The other cylinder specimens were stored in water at 20 °C to observe 7, 28, and 91 days of compressive strength under standard conditions.

The modulus of elasticity of the concrete was also measured on the cylinder specimens used for the compressive strength tests.

### 3.2. Autogenous and Drying Shrinkage

Autogenous shrinkage, identified as microscopic shrinkage, happens in concrete after the initial setting because of cement hydration [17]. Concrete has drying shrinkage as a result of the different levels of moisture inside and outside the concrete [18]. Embedded strain gauges connected to a data logger were positioned in the center of beam concrete

specimens sized 100 mm × 100 mm × 400 mm, and they were used to measure concrete strain and temperature during the experiments. Figure 2 shows the cross and side sections in the mold. Immediately after casting, the specimens were stored in a temperature-controlled room at 20 °C and 80% R.H. Then, after 2 h, they were subjected to the steam-curing protocol, as shown in Figure 1. All specimens were demolded after 24 h. The accelerated cured specimens were then subjected to drying conditions in a temperature-controlled room at 20 °C and 60% R.H. The strain and temperature were measured in all specimens up to 230 days.

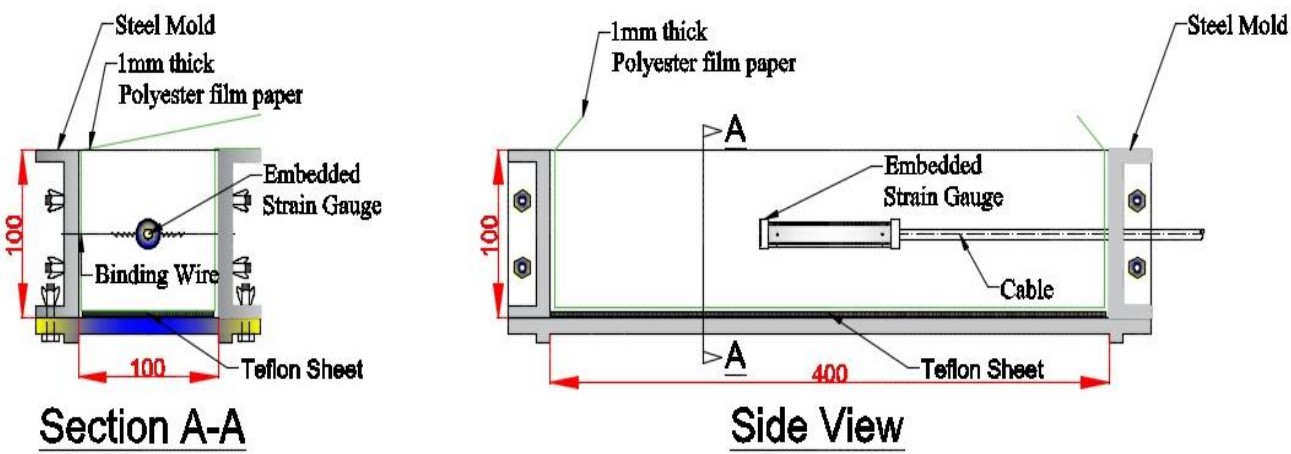

**Figure 2.** Steel mold for the concrete specimen with a strain gauge in the center.

The initial setting time, which is defined by the JCI committee [19] as the starting point of autogenous shrinkage, was 3.5 h after casting. The concrete's thermal expansion coefficient was assumed to be $10 \times 10^{-6}/°C$ in accordance with JCI (2016) [20], and autogenous shrinkage was determined by subtracting the thermal strain from the total strain.

### 3.3. Adiabatic Temperature Rise

To characterize the adiabatic temperature rise properties of concrete for mix proportions (A + FA), an adiabatic temperature rise test was conducted using an adiabatic temperature rise testing machine that used air and water circulation to control the temperature inside the chamber. Concrete was cast in a mold with a diameter of 400 mm and a height of 400 mm and was put in the temperature-controlled chamber, then an automatic controller was used to control the air temperature in the chamber to be the same as the temperature inside the concrete specimen during the same time. As the temperature of the tested concrete rises, the air temperature inside the chamber continues to rise, and the temperature of the water circulating inside the chamber also rises following the air temperature inside the chamber. The temperature of the central portion of concrete was measured for two weeks using copper-constantan thermocouples.

### 3.4. Thermal Stress Analysis Using Three-Dimensional Finite Element Method (3D FEM)

During the cement hydration process at an early concrete age, thermal cracking occurs due to restrained temperature deformations caused by excessive temperature differences within a massive concrete member of the structure or outer restraint from other attached structural members [21]. This thermal difference generates tensile stresses in concrete. As a rule, when external restraint is predominant, cracks penetrating through a concrete section (through cracks) are formed [18]. Guidelines for control of the cracking of mass concrete JCI (2016) highlight some other factors that lead to thermal cracking in concrete structures, including volume change due to the heat of cement hydration, autogenous shrinkage, and combined effects of the type of structure, boundary conditions, materials, mixture proportions, construction method, weather conditions, etc. The above factors are prominent in mass concrete structures. However, in thin concrete members, the effects of

the heat of cement hydration are low due to the insignificant temperature gradients that enable easy and uniform heat dissipation into the surrounding areas.

The JCI guidelines provide an indication of concrete behavior, used as an index of cracking probability—the thermal cracking index—which is the percentage of tensile strength from the maximum principal stresses. The chance of cracking is higher when the thermal cracking index is low. When the cracking index value is 1.0, the probability of cracking is 50%, and when it decreases to be less than 0.6, the cracking probability will increase to be 100%. Equation (1) was used to calculate the thermal cracking index [20]:

$$I_{cr} = \frac{f_t(t_e)}{\sigma_{\max}(t_e)} \tag{1}$$

where *Icr*: thermal cracking index, *ft(te)*: splitting tensile strength (N/mm$^2$), *σmax(te)*: maximum principal stress (N/mm$^2$), and *te*: temperature-adjusted age (day).

In this study, the thermal cracking index, *Icr*, includes the influence of thermal, autogenous, and drying shrinkage in different ratios; regarding that, herein, it will be referred to as the cracking index.

In this study, thermal analysis and thermal stress analysis were conducted by 3D FEM for a precast concrete product for the mix proportions (N 45%) and (A + FA 45%) using the modified fly ash cement proposed in this study, which is subjected to steam curing (which is shown in Figure 1) and then to air-curing conditions for 6 months (at 20 °C and 60% R.H.).

For FEM analysis, a precast concrete model similar to a standard precast box culvert was selected from JIS A 5372-2016—precast reinforced concrete products [22]. Figure 3 shows the actual design of a culvert box and quarter portion for numerical analysis. A computer program of thermal stress analysis for mass concrete structures (JCMAC-3) proposed by JCI, which is a 3D finite element method (FEM) simulation tool, was used. In order to simulate the time-dependent distributions of temperature and relative humidity in a concrete member, the transient heat transfer and moisture transfer analyses based on diffusion equations were carried out by 3D FEM analysis. In thermal stress analyses, autogenous shrinkage strain was added to thermal strain, and linear elastic analyses were carried out, in which stress relaxation due to creep of concrete was taken into account by using the reduction coefficient of elastic modulus, *φ(te)*, as shown below. This method is proposed by JCI guidelines.

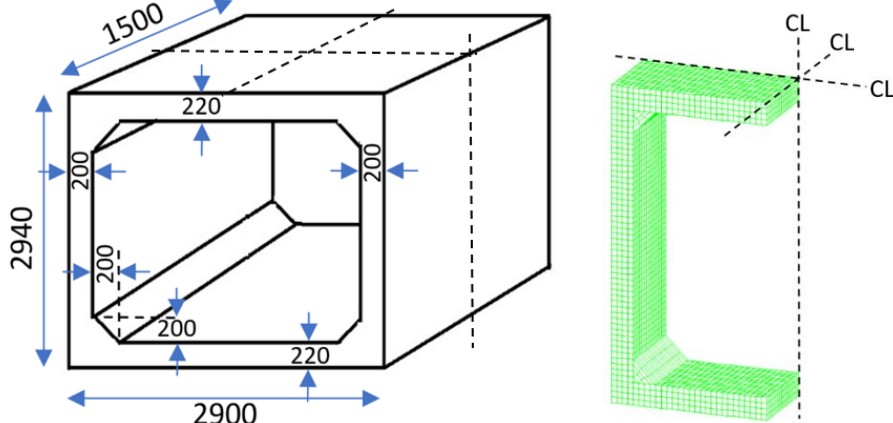

**Figure 3.** The actual dimensions of the box culvert and quarter portion. Drawing not to scale (dimensions in millimeters).

Dr. Ishikawa proposed a drying shrinkage model using the capillary tension theory for the pore size distribution of the hardened cement paste to calculate the drying shrinkage strain of the unrestrained concrete element from its relative humidity [23]. In that model, a coefficient to consider the influence of the materials used and the concrete mix proportion

on drying shrinkage was introduced. In this study, the value of the coefficient for (N 45%) was determined so that the calculated restraint stress coincided with the observation of the stress release method, where the restraint stress of a concrete specimen dried for one month was measured by measuring the elastic deformation caused by cutting off a part of the specimen [24]. The drying shrinkage strain of the concrete specimens was also measured, and the coefficient for mix proportion (A + FA 45%) was decided as the coefficient for mix proportion (N 45%) multiplied by the ratio of the drying strain of (A + FA 45%) to that of (N 45%).

In the FEM analysis, the real properties of the concrete, which were obtained as experimental results, such as adiabatic temperature rise, compressive strength, modulus of elasticity, autogenous and drying shrinkage, and casting temperature, were used. The splitting tensile strength was calculated utilizing compressive strength data and constants, as specified in the JCI guidelines for Equation (2) [20]:

$$f_t(t_e) = C_1 \times f'_C(t_e)^{C_2} \tag{2}$$

where $ft(te)$: the splitting tensile strength of concrete at $(te)$ (N/mm$^2$), $fc(te)$: the compressive strength of concrete at $(te)$ (N/mm$^2$), and $te$: temperature-adjusted age (days), $C_1 = 0.13$ and $C_2 = 0.85$.

Some other properties were also obtained from the JCI guidelines [20], such as specific heat (1.15 J/g °C) and the coefficient of thermal conductivity (2.7 w/m °C). Poisson's ratio was 0.23, and the creep of concrete's influence was evaluated using the effective modulus of elasticity, which was obtained by multiplying the modulus of elasticity by a reduction coefficient using Equation (3):

$$E_e(t_e) = \varphi(t_e) \times E_C(t_e) \tag{3}$$

where $Ee(te)$: the effective modulus of elasticity of concrete at $(te)$, $\varphi(te)$: the reduction coefficient for the modulus of elasticity due to creep, and $Ec(te)$: the modulus of elasticity of concrete at $(te)$.

At the early age during the hardening process until reaching the maximum temperature, the reduction constant, $\varphi(te)$, was taken to be 0.42, and then it increased during one day of reaching the maximum temperature to be 0.65, as a stable value for the later ages, as recommended by the JCI guidelines [20].

For Equations (1)–(3), the temperature-adjusted age $(te)$ can be calculated using Equation (4), provided by JCI [20]:

$$t_e = \sum_{i=1}^{n} \Delta t_i \times exp\left[13.65 - \frac{4000}{273 + T(\Delta t_i)/T_0}\right] \tag{4}$$

where $\Delta t_i$: the period of constant temperature continuing in concrete (day), $T(\Delta t_i)$: the concrete temperature for $\Delta t_i$ (°C), and $T_0$ is 1 °C.

As previously mentioned, the FEM stress analyses were conducted according to the procedures proposed by the JCI guidelines, including mesh generation and the material properties model. The cracking index (Equation (1)), which was calculated in accordance with the JCI guidelines, had good correlation with the cracking probability obtained from the observations for more than 728 actual mass concrete members. This can be said to demonstrate the validity of the analysis results.

## 4. Results and Discussions

### 4.1. Properties of Concrete

#### 4.1.1. Fresh Properties

Table 5 shows the results of the fresh concrete tests conducted on all mix proportions to determine their suitability for casting and compaction, including the concrete slump, temperature, and air content.

**Table 5.** Fresh properties of concrete.

| Proportion | W/B (%) | Fresh Properties | | |
|---|---|---|---|---|
| | | Slump (cm) | Air Content (%) | Temperature at Casting (°C) |
| N 45% | 45 | 12.2 | 5.4 | 18.0 |
| A + FA 45% | | 11.4 | 5.2 | 18.5 |
| N 33% | 33 | 13.1 | 4.7 | 18.5 |
| A + FA 33% | | 11.6 | 4.3 | 18.0 |

(A) high alite cement, (N) ordinary Portland cement (OPC), and (FA) fly ash.

The concrete casting temperatures were used as one of the input parameters in the 3D FEM thermal stress analysis.

4.1.2. Compressive Strength and Modulus of Elasticity

Figure 4 shows the temperatures of the specimens during the curing time. Two specimens (N-1 and N-2) for concrete with (N) and two specimens (A + FA-1 and A + FA-2) for concrete with (A + FA) were tested for each water-to-cement ratio. The profile is similar to the target temperature profile in the main plan in Figure 1.

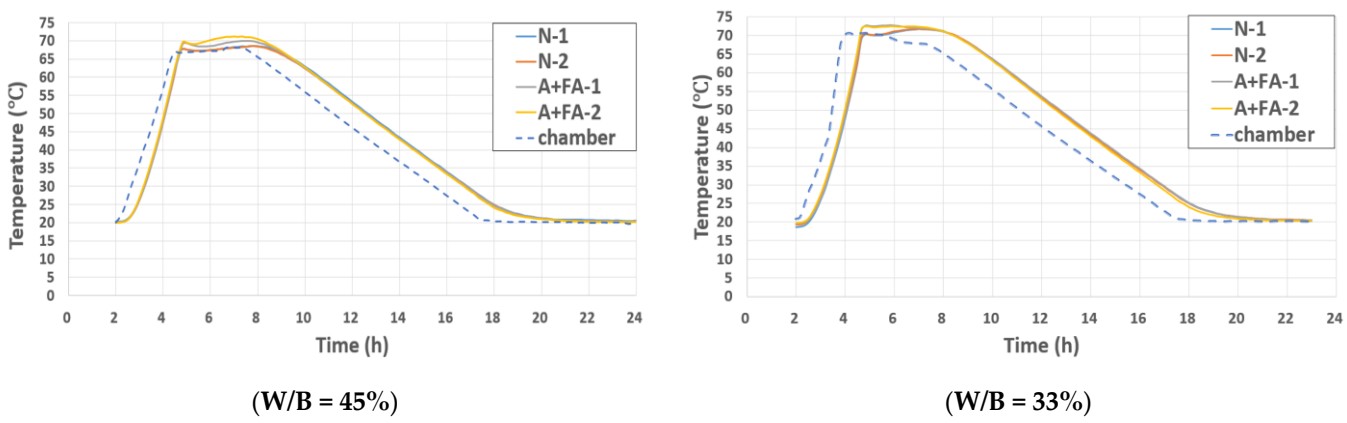

(W/B = 45%)                  (W/B = 33%)

**Figure 4.** Temperature history of steam curing.

It can be seen from Figure 5 that steam-cured concrete with fly ash and high alite cement developed higher compressive strength at one day than concrete with ordinary Portland cement. On the other hand, underwater cured concrete with (N) always showed higher compressive strength than (A + FA).

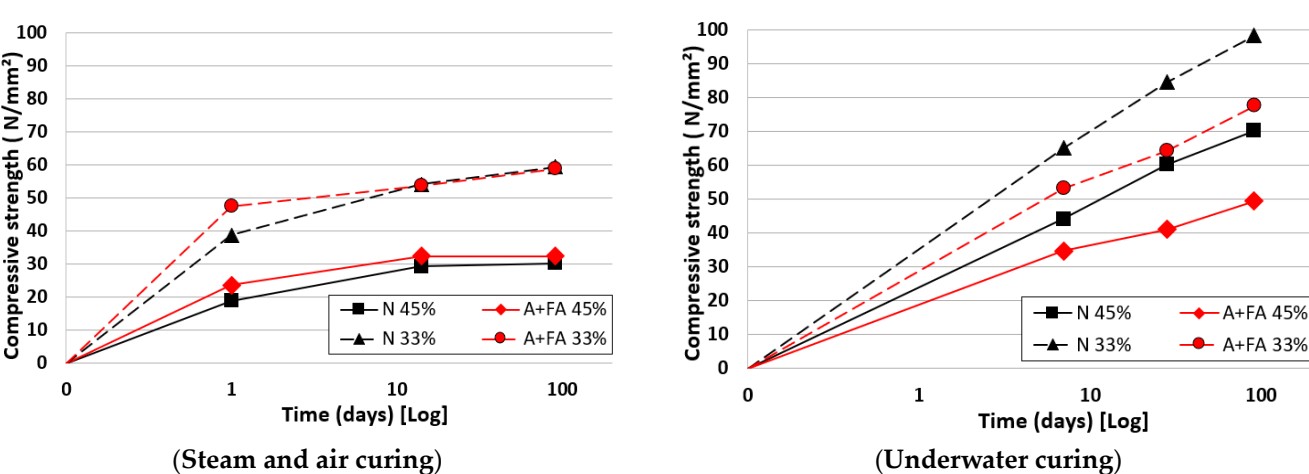

(Steam and air curing)            (Underwater curing)

**Figure 5.** Compressive strength.

For precast concrete factories, the early strength development of the proposed FA cement is highly desirable for the early demolding capability. This target was achieved by steam curing, as shown in Figure 5. However, compressive strength development was very slow after 14 days of age.

In general, the compressive strength results after the first day were much higher with underwater curing than with steam curing.

Figure 6 shows the results of the modulus of elasticity compared with the JCI model [25] and AIJ model [26], and the results were very close to the AIJ model. In most cases, the modulus of elasticity of mix proportions with (N) and (A + FA) were similar in relation to the compressive strength results.

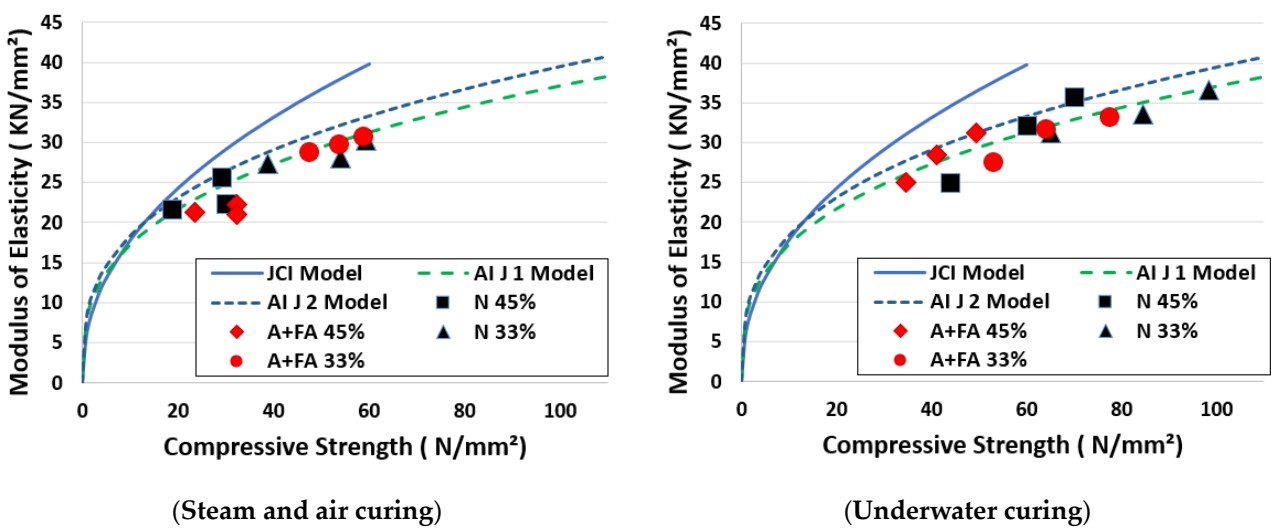

(**Steam and air curing**)    (**Underwater curing**)

**Figure 6.** Modulus of elasticity.

4.1.3. Autogenous and Drying Shrinkage

Figure 7 shows the strain due to the autogenous volume change during steam curing. It can be seen from Figure 7 that mix proportions with (N) showed higher expansion at an early age than mix proportions with (A + FA). That expansion may include thermal expansion not only due to the hydration process but also due to an increase in the surrounding temperature for the steam-curing procedure since the thermal expansion coefficient of concrete at a very early age is probably much higher than $10 \times 10^{-6}/°C$, which is an assumed value in this study. It is recommended to further study the precise estimation of the thermal expansion coefficient at very early ages.

After expansion at the very early age, concrete with a W/B of 33% showed autogenous shrinkage of around $100 \times 10^{-6}$ at the age of 24 h, which is larger than concrete with a W/B of 45%.

The drying shrinkage of concrete after steam curing is also shown in Figure 8. For concrete with a W/B = 45%, the mix proportion with (N) showed larger drying shrinkage than that with (A + FA). However, with mix proportions of W/B = 33%, (A + FA) showed larger drying shrinkage than (N), but the difference was small. These experimental results were similar to previous reports showing that drying shrinkage of concrete does not change or only slightly decreases with the addition of fly ash. Additionally, it can be seen that for the mix proportion (A + FA), the difference in drying shrinkage was very small between W/B = 33% and W/B = 45%.

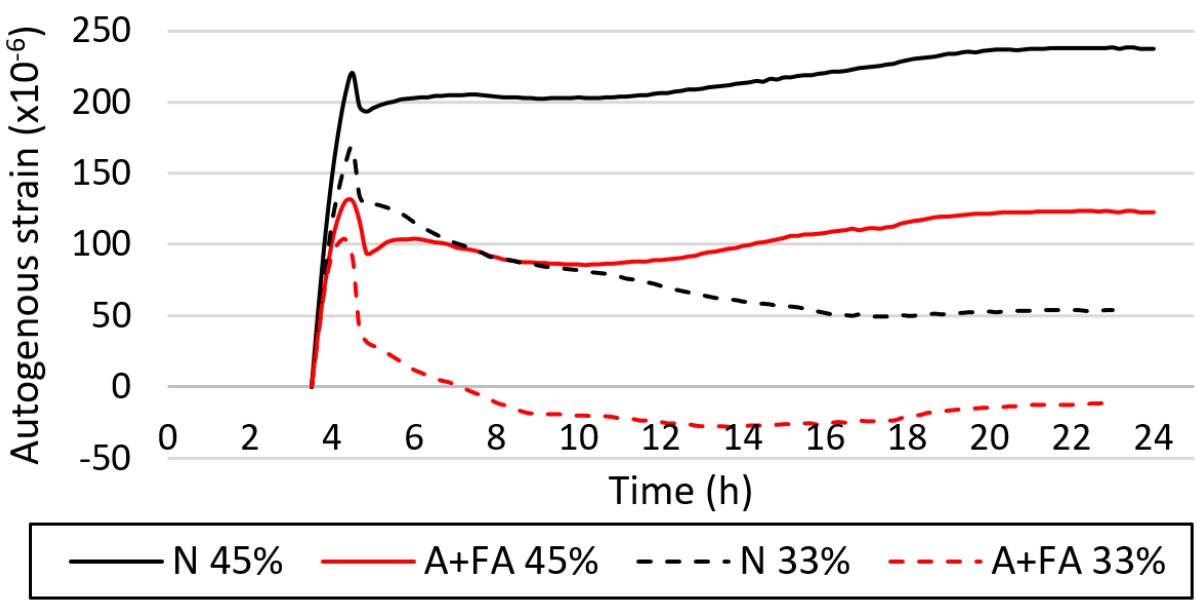

**Figure 7.** Autogenous shrinkage.

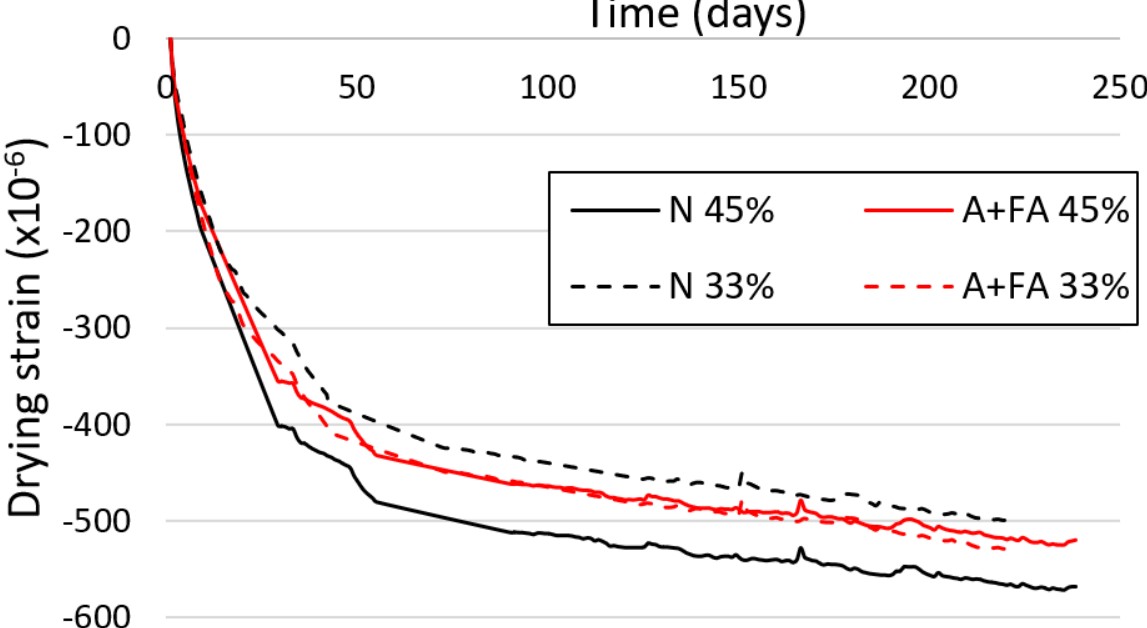

**Figure 8.** Drying shrinkage.

4.1.4. Adiabatic Temperature Rise

Figure 9 shows the results of the adiabatic temperature rise test for all mix proportions. For mix proportions (N 45%) and (N 33%), the ultimate adiabatic temperature rise was predicted from Equation (5), where $Q\infty$ and other constants that are related to the rate of temperature rise were also determined as functions of the cement content, casting temperature, and type of cement, in accordance with the JCI guidelines for control of the cracking of mass concrete (2016) [20]:

$$Q(t) = Q_\infty \left[ 1 - \exp\left\{ -r_{AT} \left( t - t_{0,Q} \right)^{S_{AT}} \right\} \right] \qquad (5)$$

where, $t$: age (day), $Q(t)$: adiabatic temperature rise at $t$ (°C), $Q_\infty$: ultimate adiabatic temperature rise (°C), $r_{AT}$, $S_{AT}$: parameters representing the rate of adiabatic temperature rise, and $t_{0,Q}$: age at the beginning of the temperature rise (day).

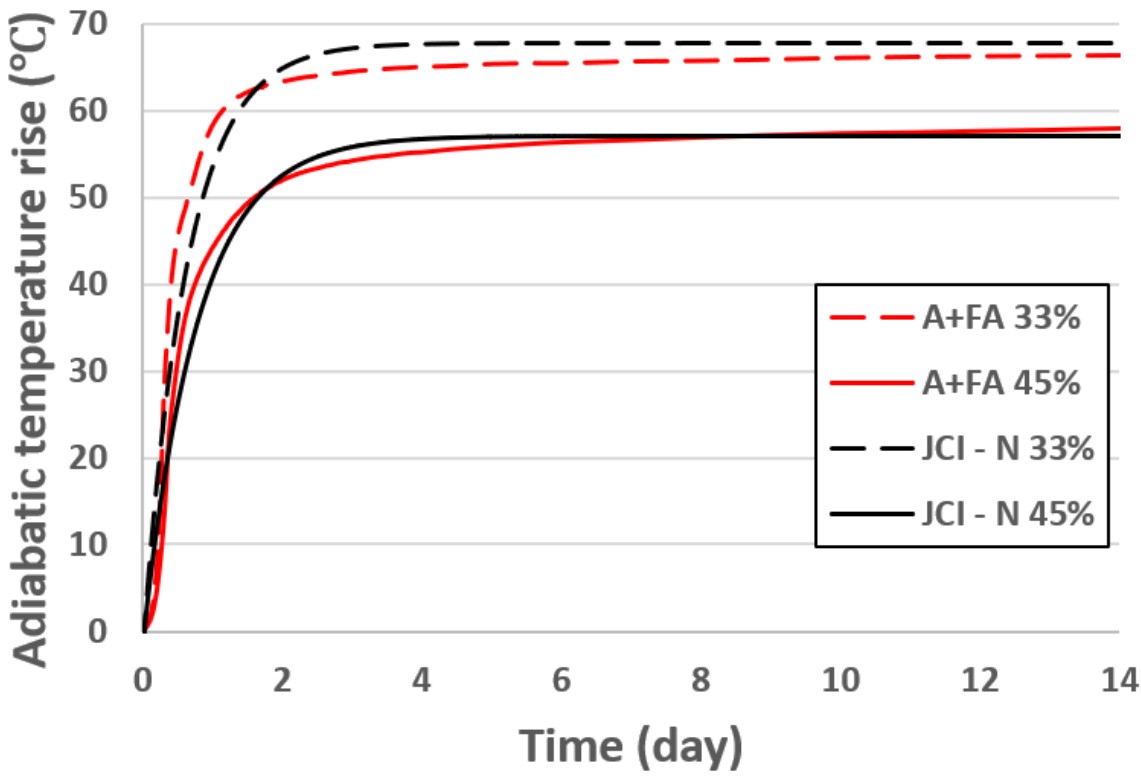

**Figure 9.** Adiabatic temperature rise due to the heat of hydration.

The temperature due to the heat of hydration was clearly higher when W/B = 33% than when W/B = 45%. The maximum temperatures were almost the same for A + FA and N with the same water-to-cement ratio. The rate of adiabatic temperature rise on the first day was higher for both A + FA mix proportions, which was related to the influence of high alite cement on hydration at early ages. Later, the rate will start to decrease for A + FA, and then it will be higher for OPC. This relation will have a great influence on the thermal stress behavior related to the type of binder, as described in Section 4.2.

*4.2. Thermal Stress Analysis*

4.2.1. Thermal Stress during Steam Curing

Using the experimental results for compressive strength, elastic modulus, adiabatic temperature rise, and autogenous shrinkage, shown in Figures 5–7 and 9 for mix proportions (N 45%) and (A + FA 45%), the FEM thermal stress analysis was conducted.

Figure 10 provides the temperature profile for points across the region at the middle of the precast concrete model for both mix proportions. The thermal analysis results showed that the maximum temperatures were in the core of the concrete member. By choosing only two points, the first at the core of the concrete element section No. 1, and the second on the surface of No. 2, it can be seen that the temperature reached its maximum value at the surface after 8 h, but for the core of the concrete element section, it reached the maximum value after 11 h from the beginning of the steam-curing process, as shown in Figure 10. Additionally, the maximum difference in temperature between the core and the surface was after 17 h from the beginning of the steam-curing process. From the analysis, the high alite cement with modified fly ash showed higher maximum temperatures than OPC due to its chemical properties.

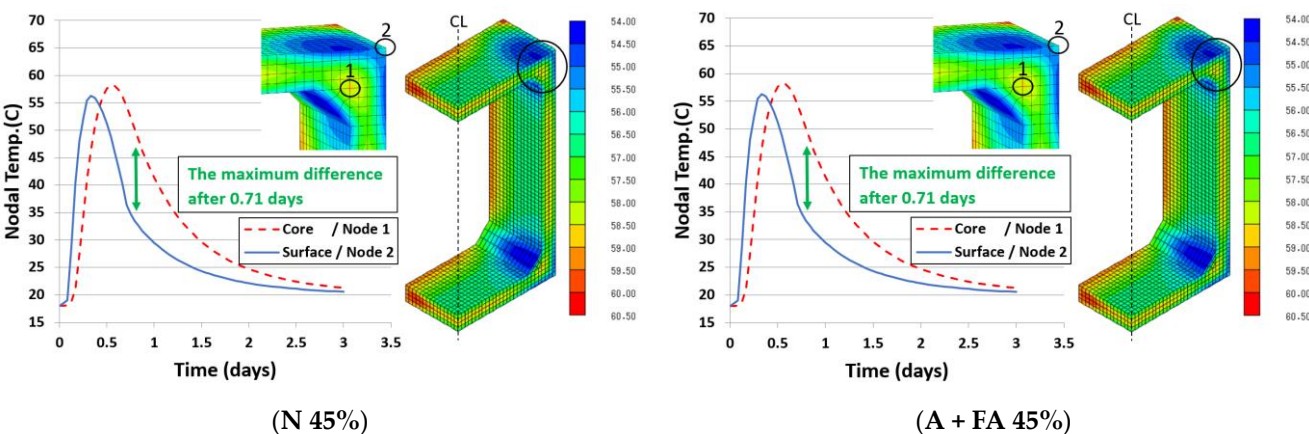

**Figure 10.** Temperature profile at the surface and the core.

Figure 11 shows the cracking index diagram considering the effects of the heat of hydration and autogenous shrinkage. The time at which the cracking index becomes the minimum value is called the critical time in that stage. It can be noticed that the critical times for both cases coincided with the time of the maximum temperature difference shown in Figure 10.

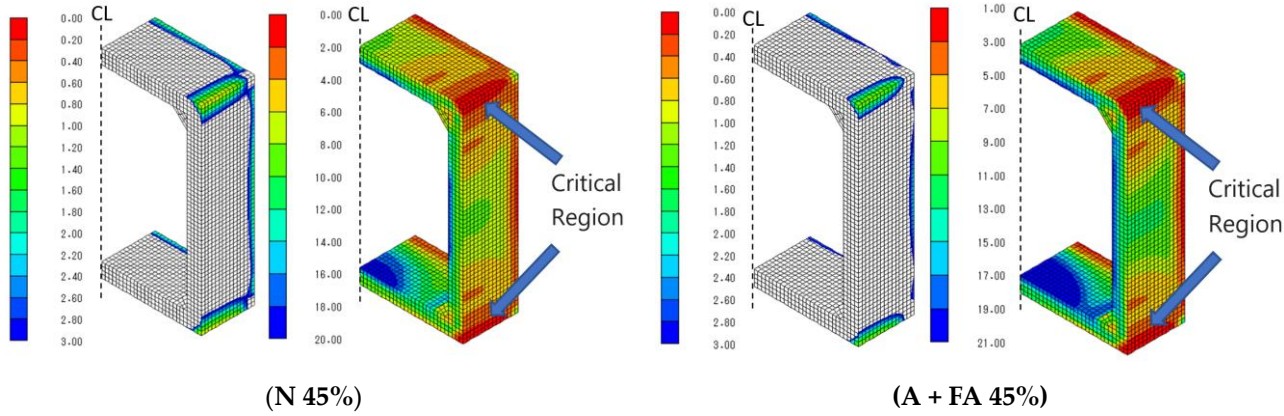

**Figure 11.** Cracking index diagram.

For the critical region, Figure 12 shows the cracking index chart by time. The cracking index was at its minimum value after 17 h from the beginning of the steam-curing process, which is the critical time in that stage. Considering the effect of autogenous shrinkage, the cracking index values were $Icr = 1.15$ and $Icr = 0.89$ for (A + FA 45%) and (N 45%), respectively. Then, after excluding the effect of autogenous shrinkage, the values were $Icr = 1.21$ and $Icr = 0.94$ for (A + FA 45%) and (N 45%), respectively. It can be said that the effect of autogenous shrinkage was very small on the cracking resistance for both mix proportions, and the contribution of autogenous strain was negligible compared with the thermal strain at this stage. For both cases, the cracking resistance of the concrete with high alite cement with fly ash was higher than that of OPC concrete.

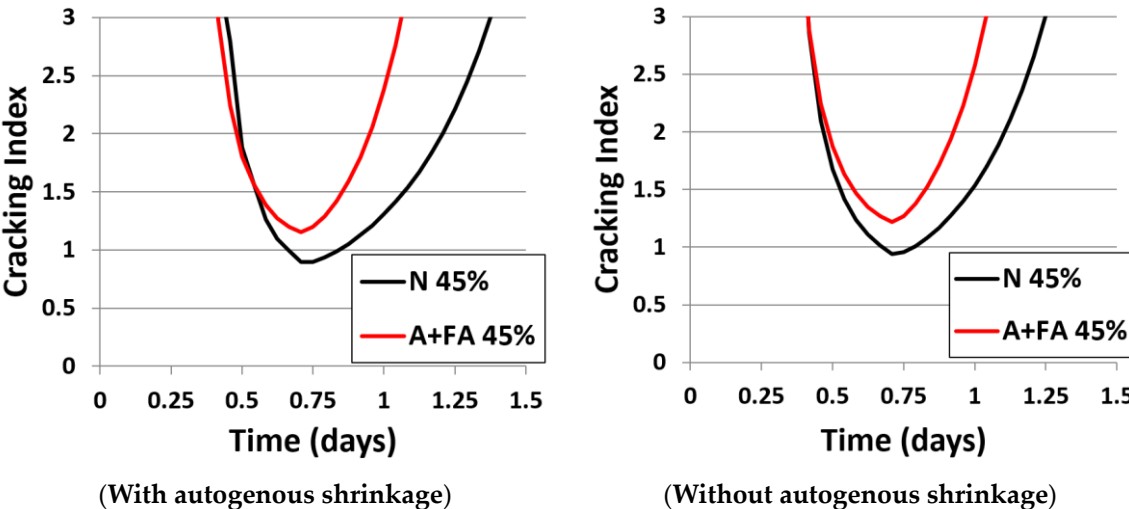

(**With autogenous shrinkage**)　　　(**Without autogenous shrinkage**)

**Figure 12.** Thermal cracking index for the critical region.

Figure 13 shows the maximum principal stresses for the critical region. The maximum values were reached after 17 h from the beginning of the steam-curing process.

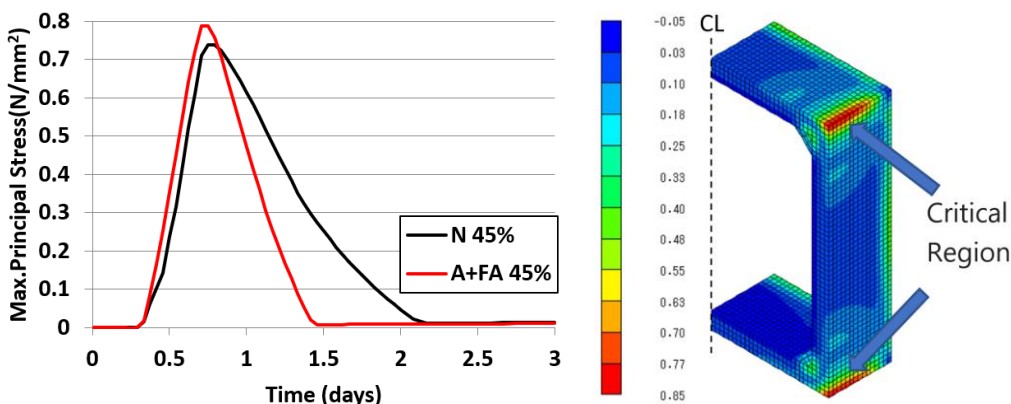

**Figure 13.** Maximum principal stresses in the critical region.

Additionally, the rate of stress decreasing for A + FA was more than that of OPC: the stresses with A + FA will be almost zero after 1.5 days, but with OPC, it will keep decreasing for more than 2 days. This is due to the increasing ratio of the adiabatic temperature rise, which is shown in Figure 9. Related to this point, it can be seen in Figure 10 that the difference in temperature between the core and the surface after 1.5 days was larger with OPC than A + FA, which means that more stresses will be generated at that time.

Comparing the diagrams for this stage, the stresses reached their maximum values at the same time when the difference between the core temperature and the surface temperature was greatest. That is, when the relative instantaneous expansion between the core and the surface of the concrete element was greatest. Therefore, internal restraint due to differential temperatures leads to the tensile stresses on the surface of the concrete element in the critical region. The directions of the maximum tensile stresses in the critical region at the critical time are provided by the FEM analysis. Therefore, cracking patterns can be predicted since the cracks will be developed perpendicular to the directions of the tensile stresses, as shown in Figure 14.

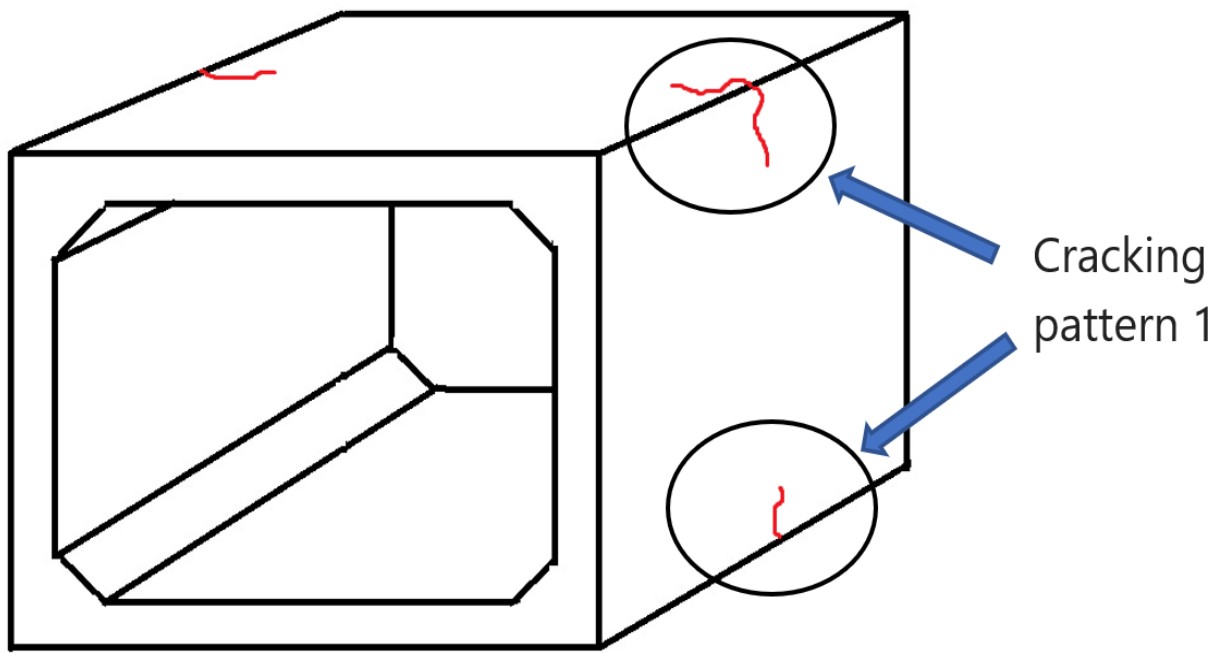

**Figure 14.** Cracking pattern in the first stage.

4.2.2. Drying Shrinkage Stress after Steam Curing

The second stage was conducted to study the effects of drying shrinkage on restraint stress generation and cracking after steam curing.

Figure 15 shows the concrete humidity changes on the outside surface and in the core of the concrete member. During this time, the relative humidity of the surface will rapidly decrease. The difference in relative humidity between the surface and the core was at its maximum value at 75 days from the beginning of the process. In general, the surface of the concrete loses moisture much faster than the internal parts of the concrete member, and internal restraint due to differential humidity will lead to the tensile stresses on the surface of the concrete.

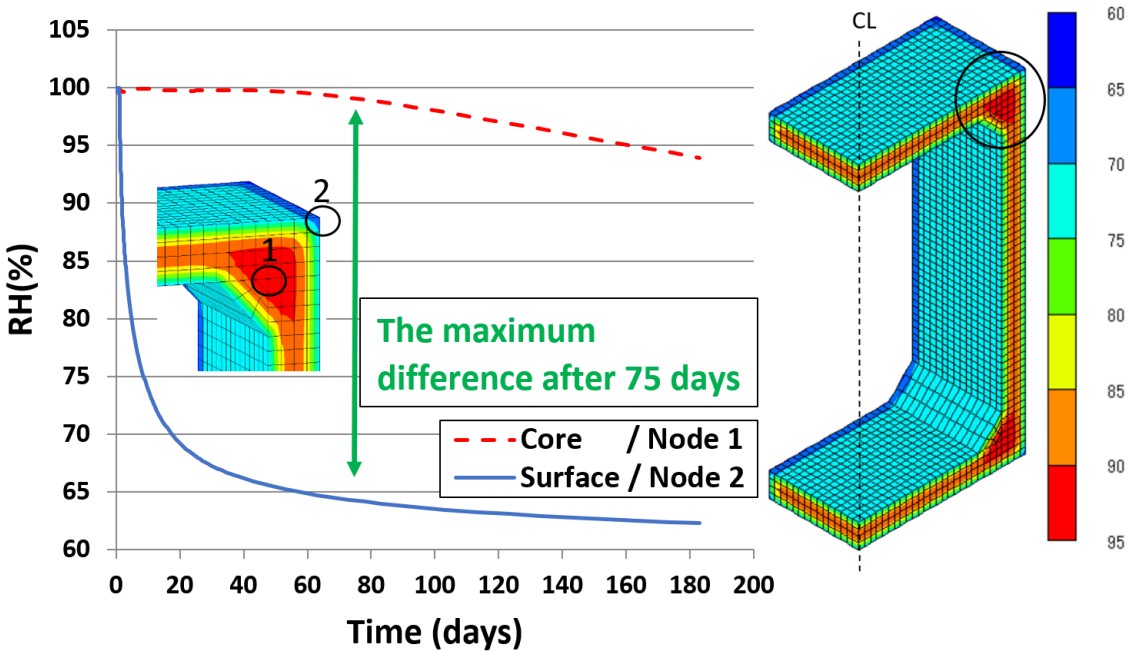

**Figure 15.** Concrete humidity (W/B = 45%).

Figure 16 shows the minimum cracking index considering the drying shrinkage, in which the critical region (1) shown in Figure 15 and another two critical regions (2 and 3) are shown.

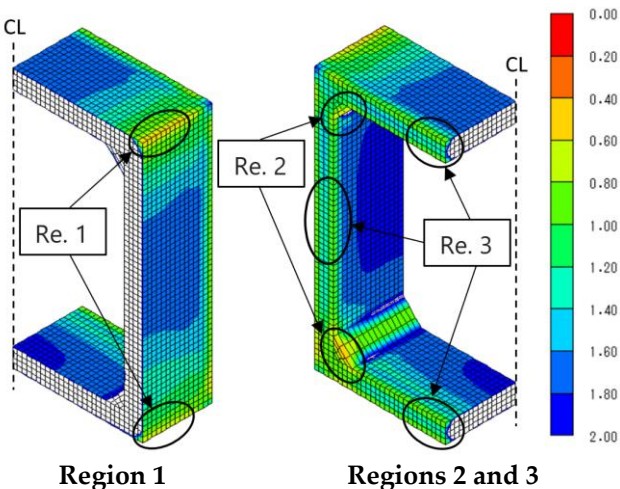

**Figure 16.** The minimum cracking index considering drying shrinkage.

Figure 17 shows the cracking index changes by time. The cracking index reached the minimum values after 180 days. Regions 1 and 2 had the same final values of $Icr = 0.63$ and $Icr = 0.0.52$ for (A + FA 45%) and (N 45%), respectively, while the values for region 3 were $Icr = 0.91$ and $Icr = 0.75$ for (A + FA 45%) and (N 45%), respectively. As a result, it can be said that the cracking resistance with high alite cement with modified fly ash was higher than it was with OPC, but the cracking probability may be very high for both mix proportions as well since the cracking index was much lower than 1.0.

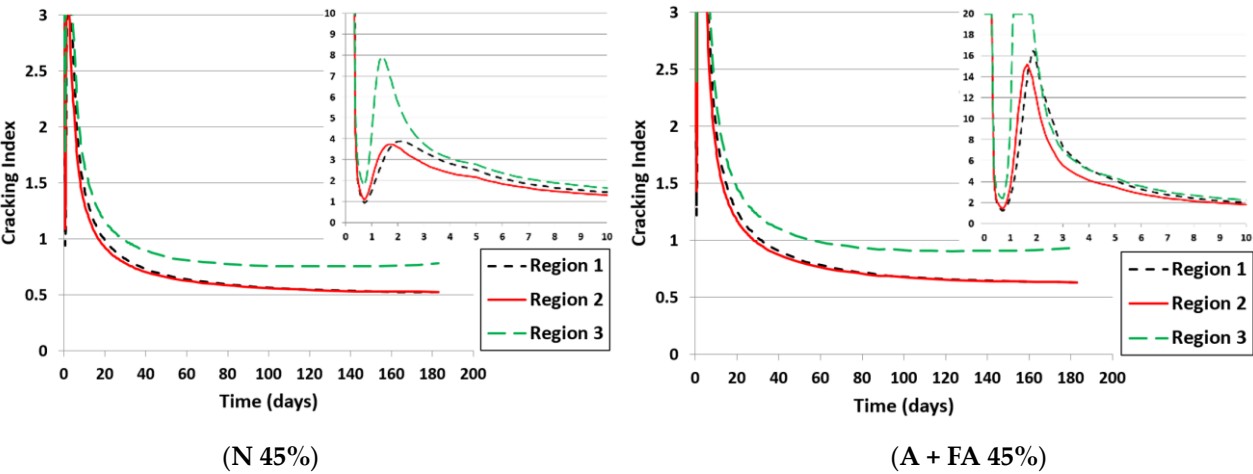

**Figure 17.** Thermal cracking index for the critical regions.

In the FEM analysis, the directions of the maximum tensile stresses in the critical regions were also obtained. Therefore, cracking patterns can be predicted since the cracks will be developed perpendicular to the directions of the tensile stresses. It was also shown from the FEM analysis that other patterns (such as 2 and 3) will appear on the outside edges of the concrete member in addition to cracking pattern 1, as shown in Figure 18.

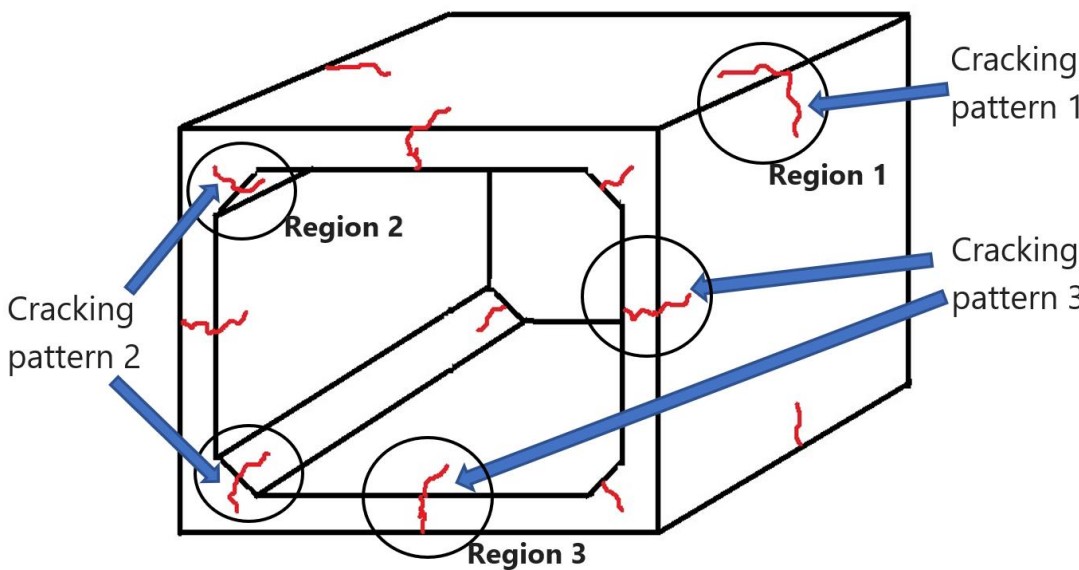

**Figure 18.** Cracking patterns in the second stage.

Table 6 presents a summary of the results for the minimum cracking index, including the results presented in Section 4.2. During steam curing, the value of the cracking index for the concrete with the proposed modified fly ash cement was more than 1.0, so it can be said that there would not be a high probability of cracking. On the other hand, the cracking index for OPC concrete was less than 1.0, so the probability of cracking was relatively higher due to thermal strain. After the steam-curing process, the values of the cracking index were much lower than 1.0 for A + FA and OPC, especially in regions 1 and 2, so it can be said that there would be a very high probability of cracking development due to drying shrinkage strain. The cracking resistance was always higher with high alite cement with modified fly ash (A + FA) than with ordinary Portland cement (N).

**Table 6.** The minimum cracking index.

| Mix Proportion | During Steam Curing | After Steam Curing up to 6 Months | | |
|:---:|:---:|:---:|:---:|:---:|
| | Region 1 | Region 1 | Region 2 | Region 3 |
| N 45% | 0.89 | 0.52 | 0.52 | 0.75 |
| A + FA 45% | 1.15 | 0.63 | 0.63 | 0.91 |

## 5. Conclusions

1. Steam-cured concrete with modified fly ash and high alite cement developed a higher compressive strength on the first day of age than concrete with ordinary Portland cement.
2. There were no big differences in the modulus of elasticity between the steam-cured concrete and the underwater-cured concrete, regardless of fly ash addition.
3. It was proven from FEM stress analysis for a steam-cured box culvert that the effect of autogenous shrinkage on the cracking probability was very small. It can be said that thermal shrinkage was the dominant factor for generating internal stresses in the concrete at the early age. On the other hand, drying shrinkage dominated at later ages due to the decrease of the internal humidity of the concrete.
4. Using the proposed high alite cement with modified fly ash improved the cracking resistance of the precast concrete box culvert more than OPC during the steam-curing process.
5. After steam curing, drying shrinkage was the main cause of cracking in the precast concrete box culvert at later ages, regardless of the type of binder.

**Author Contributions:** S.M., A.A., N.N., R.K. and E.S. conceived and designed the experiments; A.A., S.M., N.N. and R.K. performed the experiments; A.A., S.M. and E.S. analyzed the data; A.A. and S.M. contributed to manuscript preparation and participated in critically revising the article for important intellectual content. All authors have read and agreed to the published version of the manuscript.

**Funding:** This research received no external funding.

**Data Availability Statement:** The data presented in this study are not available.

**Acknowledgments:** The study was carried out as an activity of the "Technical Committee on AFC for Precast Concrete". The authors express their sincere gratitude to the parties concerned. Additionally, the first author expresses his great gratitude to the Japan International Corporation Agency (JICA), the JELA foundation, and The Kubota Fund, for supporting him during his journey.

**Conflicts of Interest:** The authors declare no conflict of interest.

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
