# Peer review of "Cracking Resistance of Steam-Cured Precast Concrete Using High Alite Cement with Modified Fly Ash"

_infrastructures, doi:10.3390/infrastructures8100154_

Round 1

Reviewer 1 Report

Dear authors,

it was my pleasure to read and review your manuscript. The results are nicely presented and I have rather minor comments, improving quality of the manuscript:

- the term “fly ash cement” evokes cement with significant fly ash replacement, such as CEM II/B, CEM III or even CEM IV. In fact, you use Portland cement and fly ash as an admixture. More appropriate term would be “cement with fly ash”. Please check those occurrences in abstract and the following text.

- Line 28 – please include some relevant reference for fly ash use in concrete, e.g. Fly Ash in Concrete. Properties and performance, 2014, or any similar book.

- Table 2. Blaine fineness of FA-1, FA-2 and FA seems to be quite high. I believe there was some preprocessing such as grinding. It would be beneficial to include source of fly ash, CaO content and reactive (amorphous) SiO2 content as well and classification as C or F type. There are many variant of fly ash and it is hard to judge possible behavior which also differs broadly.

- Need to fix capital letter in Table 4 title, Section 3.2. title, line 173, 191, Section 4.1.2. title.

- Line 193 “decrees” should be likely “decreases”

- Line 214. “Linear elastic analysis” can never give stress relaxation. I think you mean “linear viscoelastic analysis” with probably rate-type approach, which is used in the majority of codes to overcome deficiencies of integral approach with memorizing full stress history.

- Line 223-224 needs more explanation what coefficients you modified and how you achieved stress release, or what you mean by that.

- Lines 246, 249 – reduction coefficient should have the same symbol as in Equation (3).

- Figure 5 – consider using log on x-axis.

- Section 4.1.3. Autogeneous shrinkage is measured from the final setting according to ASTM C1698 − 19. This could help to set zero shrinkage later, the first two curves look just shifted. There should be no reason for such a big expansion, young concrete has higher coefficient thermal expansion coefficient which could help to reduce the autogeneous strain. Please put on y-label “Autogenous strain (x10-6)” in Figure 7.

- Line 301 shrinkage has no degree Celsius in units.

- Figure 9, misspelled Temperature. Better y-axis label would be “Adiabatic temperature rise (°C)”.

Author Response

Dear respected reviewer

It will be our pleasure to work with you.

Reviewer 2 Report

In this article, the effect of the new fly ash cement with both high alite (C3S) cement and fly ash modified by electrostatic belt separation method on cracking resistance of precast concrete prepared by steam curing was studied. Some interesting conclusions were got. In general, this article is well done and has some innovations.

There are the following questions and suggestions:

1. In Table 2, three materials including FA-1, FA-2 and FA were listed. When are they used? What is the difference between FA and FA-1 or FA-2?

2. In Figure 4, what materials do N-1 and N-2 refer to?

3. In section 4.2, the comparation between N 45% and A+FA 45% is conducted. Why is not the comparation between N 33% and A+FA 33% done?

4. Why not to validate the models used in section 4.2?

5. Figure 17 is too wide to fit the page.

6. The third conclusion needs some explanations.

 In general, this article is well written. Only minor editing of English language is required.  For example, the capitalization of some words are wrong. They include but are not limited to:

   (1) “Ash” in line 105;

(2) “Aggregates” in line 113;

(3) “when” in line 192;

(4) “T0 “in line 255.

Author Response

(The authors gave the same response as above.)

Reviewer 3 Report

Congratulations to the authors on their interesting work.

The use of cements with the addition of fly ash is certainly a desirable direction in line with the circular economy and is an interesting alternative to Portland cements without additives.

The results obtained during the experiments confirm the assumptions and theses of the work.

In my opinion, the value of the work can be increased by supplementing several results, therefore:

1. I am asking authors to provide information regarding possible reduction of the carbon footprint

2. I am asking the authors to complete the information regarding the durability of concrete using the above-mentioned cements.

Author Response

(The authors gave the same response as above.)
